# Central Venous Catheters versus Peripherally Inserted Central Catheters: A Comparison of Indwelling Time Resulting in Colonization by Multidrug-Resistant Pathogens

**DOI:** 10.3390/antibiotics13010089

**Published:** 2024-01-17

**Authors:** Vassiliki C. Pitiriga, John Bakalis, Elsa Campos, Petros Kanellopoulos, Konstantinos Sagris, George Saroglou, Athanasios Tsakris

**Affiliations:** 1Department of Microbiology, Medical School, National and Kapodistrian University of Athens, 75 Mikras Asias Street, 11527 Athens, Greece; vpitiriga@med.uoa.gr; 2Department of Internal Medicine, Metropolitan Hospital, 9 Ethnarchou Makariou Street, 18547 Athens, Greece; jhnbakalis@gmail.com (J.B.); espiro517@yahoo.com (E.C.); petrokan@otenet.gr (P.K.); ksagris@gmail.com (K.S.); gs200744@otenet.gr (G.S.)

**Keywords:** catheterization, central venous catheter, colonization, peripherally inserted central catheter, catheter duration, indwelling time

## Abstract

Background: The use of peripherally inserted central catheters (PICCs) as an alternative to central venous catheters (CVCs) has steadily risen over the last two decades. However, there is an ongoing debate regarding research evidence that supports any clear advantages or disadvantages of them compared to traditional central venous lines. The present study was conducted to compare the indwelling time of CVC and PICC placements leading to microbial colonization by multidrug-resistant microorganisms (MDROs) in critically ill patients. Methods: A single-center retrospective descriptive study was performed that reviewed the medical records of critically ill patients with colonized CVCs and PICCs who were hospitalized during a 24-month period (May 2019–May 2021). To evaluate the association between indwelling time of catheter placement and colonization rates, events were categorized into three groups, each representing a one-week time interval of catheter indwelling time: group 1: ≤7 days, group 2: 8–14 days, and group 3: >14 days. Results: A total of 207 hospitalized patients with colonized PICCs or CVCs were included in the study. Of these, 144 (69.5%) had a CVC placement and 63 (30.5%) had a PICC placement. The overall colonization rate (per 1.000 catheter/days) was 14.73 in the CVC and 5.67 in the PICC cohort (*p* = 0.003). In the group of PICCs, 12/63 (19%) of the pathogens were MDROs and 51/63 (81%) were non-MDROs, while in the group of CVCs, 86/144 (59.7%) were MDROs and 58/144 (40.3%) were non-MDROs (*p* < 0.001). The colonization rate in the CVC cohort, was 6.98 for group 1, 21.57 for group 2, and 21.6 for group 3 (*p* = 0.019). The colonization rate of MDROs was 3.27 for group 1, 14.47 for group 2, and 12.96 for group 3 (*p* = 0.025). Regarding the PICC cohort, the colonization rate was 1.49 for group 1, 3.19 for group 2, and 8.99 for group 3 (*p* = 0.047). No significant difference existed between the three groups in terms of MDRO pathogens, with the colonization rate being 0 for group 1, 0.8 for group 2, and 1.69 for group 3 (*p* = 0.78). Within the CVC cohort, the most common isolated microorganism was MDR *Acinetobacter baumannii* (*n* = 44; 30.6%), followed by MDR *Klebsiella pneumoniae* (*n* = 27; 18.7%). In the PICC cohort, the predominant isolated microorganism was *Candida non-albicans* (*n* = 15; 23.8%), followed by *Candida albicans*, coagulase-negative staphylococci, and MDR *Klebsiella pneumoniae* in equal numbers (*n* = 6; 9.5%). Conclusions: Our findings show that while the indwelling time of PICC placement was longer compared to CVCs, its colonization rate was considerably lower. Furthermore, high colonization rates by microorganisms, especially MDROs, arose later during catheterization in PICCs compared to CVCs, suggesting that in terms of vascular infections, PICCs may be a safer alternative to conventional CVCs for long-term intravenous access.

## 1. Introduction 

Central venous catheters (CVCs) and peripherally inserted central venous catheters (PICCs) play a crucial role in daily medical practice, particularly in intensive care units (ICUs). CVCs are frequently employed in the management of severely ill hospitalized patients, facilitating the administration of therapeutic and supportive treatments when dealing with challenging venous access or the need for prolonged or frequent infusions. Commonly used devices include short-term CVCs, placed in the internal jugular, femoral, or subclavian vein, medium-term catheters, like PICCs, and long-term tunneled CVCs with fully implanted ports [1]. 

The utilization of PICCs as an alternative to CVC access has gradually increased over the past two decades, primarily due to the straightforward insertion and removal procedures, enhanced safety, potential cost-effective benefits, and their suitability for intravenous therapy in home care patients [2,3,4]. Despite their widespread use, numerous studies have examined whether PICCs offer advantages or disadvantages compared to CVCs in routine medical practice, particularly concerning potential complications [5,6]. However, apart from a tendency toward a higher incidence of thrombotic cases compared to implantable devices [7] and a reduced risk for catheter occlusion with PICCs [8], research evidence supporting any clear advantage or disadvantage of PICCs over traditional central venous lines is still debated [9,10]. Specifically regarding the risk for catheter-related infections and sepsis, although there is support for the view that PICCs are safer than CVCs, there is insufficient evidence to routinely recommend a specific type of CVC over the other [11,12]. Significant discrepancies exist in the reported rates of central line-associated bloodstream infections (CLABSIs) among studies, while limited data exist regarding the distinct effects of risk factors associated with each catheter type, including variables such as the site of catheter insertion [13], the duration of catheterization [14], patient comorbidities, and catheter colonization [15]. 

Our recently published study [16] regarding the existing research evidence for catheter colonization of CVCs and PICCs has reported that PICC lines exhibited significantly lower colonization rates and a lower prevalence of multidrug-resistant (MDR) Gram-negative organisms compared to CVCs in critically ill patients. Following this, we conducted the present study to further examine the differences in the indwelling time of CVC and PICC placements leading to microbial colonization by multidrug-resistant microorganisms (MDROs) in critically ill patients. We compared colonization rates between CVCs and PICCs at three different time intervals during catheter placements. Additionally, our goal was to provide information about the distribution and species identification of common pathogens and MDROs across distinct time intervals.

## 2. Materials and Methods

We conducted a retrospective analysis of data obtained from consecutive admissions of critically ill patients to Metropolitan Hospital, a tertiary care private hospital in Athens, Greece. The data cover a 24-month period from May 2019 to May 2021. To assess the relationship between the indwelling time of catheter placement and colonization rates, we categorized events into three groups, each representing a 1-week time interval: group 1 (≤7 days), group 2 (8–14 days), and group 3 (>14 days). This observational study received approval from the hospital’s institutional review board.

### 2.1. Data Collection

After insertion, catheters were checked using a check-box form containing the patient’s diagnosis, operator’s name, site chosen, date placed and removed, date of ICU discharge or death, mechanical ventilation, arterial catheters, parenteral nutrition, vasopressor support, and daily clinical assessment (e.g., induration, discharge, erythema, and tenderness) of possible catheter infection. The operator inserting the catheter entered the initial data, nurse personnel entered data the following days, and the infection control nurse monitored data collection 3–4 times per week. We retrospectively collected study data from three different data sources: (1) the ICU database (for demographic and clinical data related to the patient’s admission and clinical course); (2) the clinical laboratory, and (3) the hospital infection control team database.

### 2.2. Catheter Care

Standardized catheter care was achieved by highly trained nursing staff proficient in all aspects of catheter care. All insertion sites were maximally visualized for potential dressing contamination. Every couple of days or earlier if clinically required, the nursing staff changed the dressing, cleaned the skin site and the catheter hub with iodine solution, and changed the intravenous accessory tubing. Furthermore, the nursing staff independently enforced a sterile insertion technique.

### 2.3. Indications for Catheter Removal

Catheters were removed under the following circumstances: (a) when there was a suspicion of infection, (b) when the catheter was no longer needed, and (c) after 15 days of insertion in the case of CVCs. The 15-day indwelling time aligns with our institutional policy aimed at preventing CVC infections arising from prolonged catheter use. This policy is based on the incidence of CVC infections from the institution’s infection control program. In certain situations, CVCs were retained beyond 15 days if the risk of acquiring new venous access outweighed the potential risks of leaving the current CVC in place. This time criterion was not applicable to PICCs, which were not removed unless the first two criteria were met.

### 2.4. Culture Techniques

All catheters were examined for the presence of pathogens either as a routine after removal or after suspicion of infection. After disinfecting skin around the catheter entry site, the proximal 4–5 cm part of the tip was cut off using sterile scissors. The specimen was placed in a sterile container and transported to the microbiology laboratory within 15 min at room temperature. The intradermal and intravascular portion of the catheter was analyzed by the semiquantitative culture technique described by Maki et al. [17]. According to Maki’s technique, catheter tip culture is considered positive in the presence of ≥15 colony-forming unit (CFU) growth of any organism. Blood cultures were incubated in Becton Dickinson Bactec (BD Bio-sciences, San Jose, CA, USA) in aerobic and anaerobic broth media. Identification of isolates and antimicrobial resistance patterns were determined by the VITEK^®^2Automated Compact System (BioMérieux Co., Lyon, France). An E-test (BioMérieux Co., Lyon, France) was performed as an additional test in order to confirm the resistance phenotypes reported by the VITEK System, according to the standard laboratory procedures. 

### 2.5. Definitions

*A CVC* was defined as any central venous access device inserted into the internal jugular, subclavian, or femoral vein that terminated in the inferior vena cava or right atrium. 

*PICCs* were defined as catheters inserted in the basilic, cephalic, or brachial veins of the upper extremities with tips that terminated in the superior vena cava or right atrium. 

*Catheter days* were defined as the number of CVCs/PICCs present among all units’ patients at 08:00 h each morning. 

*Multidrug-resistant organisms (MDROs)* were defined as the species of microorganisms that exhibit antimicrobial resistance to at least one antimicrobial drug in three or more antimicrobial categories [18]. This definition concerns both Gram-positive and Gram-negative bacteria.

*Catheter colonization* was considered by the presence of a semi-quantitative culture of ≥15 CFU of at least a single organism per catheter, according to Maki et al. [14].

### 2.6. Statistical Analysis

Descriptive analyses to characterize the patient population were reported as count (percent) or mean value (+/− standard deviation) for qualitative and quantitative variables, respectively. Comparisons between the three groups were carried out using chi-square, an independent samples *t*-test, or a one-way ANOVA test, as appropriate. A *p*-value of <0.05 was considered statistically significant.

## 3. Results

### 3.1. Participants Characteristics 

A total of 207 hospitalized patients with colonized PICCs and CVCs were included in the study. Of them, 144 (69.5%) had a CVC placement and 63 (30,5%) had a PICC placement. The mean indwelling time of catheterization was 14.05 ± 9.7 days for CVCs (range: 1–55 days) and 26.67 ± 10.6 days for PICCs (range: 5–72 days) (*t*-test, *p* = 0.001). The total patients’ demographic and clinical characteristics are presented in Table 1. No differences in demographic characteristics were determined between the two groups (Table 1). Also, no differences existed in the proportion of catheterization sites—femoral, internal jugular, and subclavian—among the three sites in the CVC cohort. 

### 3.2. Colonization Incidence Rates and MDROs/Non-MDROs Proportion in the CVC and PICC Cohorts

A significantly higher colonization rate was determined in CVCs compared to PICCs. Specifically, the overall colonization rate (per 1.000 catheter/days) was 14.73 in the CVC cohort and 5.67 in the PICC cohort (*t*-test, *p* = 0.003). Among all microorganisms isolated from both PICCs and CVCs, 109/207 (52.7%) were non-MDROs and 98/207 (47.3%) were MDROs. A significantly higher proportion of MDROs were isolated from CVCs compared to PICCs. More specifically, in the cohort of PICCs, 12/63 (19%) of the pathogens were MDROs and 51/63 (81%) were non-MDROs, while in the cohort of CVCs, 86/144 (59.7%) were MDROs and 58/144 (40.3%) were non-MDROs (*X*^2^, *p* < 0.001).

### 3.3. Colonization Incidence Rates among the Three Groups in the CVC Cohort

Within the CVC cohort, the colonization rate was significantly higher in groups 2 and 3 than in group 1. Specifically, the colonization rate was 6.98 per 1.000 catheter/days for group 1, 21.57 per 1.000 catheter/days for group 2, and 21.60 per 1.000 catheter/days for group 3 (one-way ANOVA, *p* = 0.019). The same trend was observed in terms of MDROs among the three groups, with groups 2 and 3 presenting significantly higher rates of colonization than group 1. More specifically, the colonization rate in group 1 was 3.27 per 1.000 catheter/days, 14.47 per 1.000 catheter/days in group 2, and 12.96 per 1.000 catheter/days in group 3 (one-way ANOVA, *p* = 0.025). In contrast, no significant difference existed between the three groups in terms of non-MDRO pathogens. Specifically, the colonization rate was 3.71 per 1.000 catheter/days for group 1, 7.31 per 1.000 catheter/days for group 2, and 8.64 per 1.000 catheter/days for group 3 (one-way ANOVA, *p* = 0.5; Table 2). 

### 3.4. Colonization Incidence Rates among the Three Groups in the PICC Cohort

Regarding the distribution of colonization among the groups of PICCs, the incidence rate was significantly higher in group 3 compared to the other two groups, i.e., 1.49 per 1.000 catheter/days for group 1, 3.19 per 1.000 catheter/days for group 2, and 8.99 per 1.000 catheter/days for group 3 (one-way ANOVA, *p* = 0.047). No significant difference existed between the three groups in terms of MDRO pathogens, with the colonization rate being 0 per 1.000 catheter/days for group 1, 0.8 per 1.000 catheter/days for group 2, and 1.69 per 1.000 catheter/days for group 3 (*t*-test, *p* = 0.78). The colonization rate due to non-MDROs among the three groups was 1.49 per 1.000 catheter/days for group 1, 2.40 per 1.000 catheter/days for group 2, and 7.30 per 1.000 catheter/days for group 3 (one-way ANOVA, *p* = 0.054) (Table 3).

### 3.5. Comparison of Colonization Events, MDROs, and Non-MDROs between the Three Groups in the CVC and PICC Cohorts 

A significant difference in the distribution of colonization events was determined between the three groups in PICCs and CVCs. More specifically, in the CVC cohort, 32/144 (22.2%) colonization events were in group 1, 62/144 (43.1%) were in group 2, and 50/144 (34.7%) were in group 3, while in the PICC cohort, 3/63 (4.8%) colonization events were in group 1, 12/63 (19%) were in group 2, and 48/63 (76.2%) were in group 3 (*X*^2^, *p* < 0.001; Figure 1a). A significant difference in the proportion of MDROs was identified between the three groups in the cohorts of PICCs and CVCs. In the CVC cohort, 15 (17.4%) cases of MDROs were isolated in group 1, 41 (47.7%) were isolated in group 2, and 30 (34.9%) were isolated in group 3. In the PICC cohort, zero (0%) cases of MDROs were isolated in group 1, three (25%) were isolated in group 2, and nine were isolated (75%) from group 3 (*X*^2^, *p* = 0.023; Figure 1b).

A statistically significant difference was also established in the proportion of non-MDROs between the three groups in the PICC and CVC cohorts. In the CVC cohort, 17/58 (29.3%) patients with non-MDROs were classified in group 1, 21/58 (36.2%) were classified in group 2, and 20/58 (34.5%) were classified in group 3, while in the PICC cohort, 3/51 (5.9%) were classified in group 1, 9/51 (17.6%) were classified in group 2, and 39/51 (76.5%) were classified in group 3 (*X*^2^, *p* < 0.001; Figure 1c).

### 3.6. Comparison of MDRO/Non-MDRO Proportions between the Three Groups in the CVC and PICC Cohorts

There was no significant difference in the proportion of MDROs to non-MDROs within each of the three groups in both the CVC and PICC cohorts. More specifically, in the PICC cohort, zero (0.0%) MDROs and three (5.9%) non-MDROs were isolated in group 1, nine (17.6%) MDROs and three (25%) non-MDROs were isolated in group 2, and thirty-nine (76.5%) MDROs and nine (75%) non-MDROs were isolated in group 3 (*X*^2^, *p* = 0.61; Figure 2). Concerning the cohort of patients with CVCs, the distribution of MDROs and non-MDROs (%) was as follows: 17 (29.3%) MDROs and 15 (17.4%) non-MDROs in group 1, 21 (36.2%) MDROs and 41 (47.7%) non-MDROs in group 2, and 20 (34.5%) MDROs and 30 (34.9%) non-MDROs in group 3 (*X*^2^, *p* = 0.196; Figure 3).

### 3.7. MDRO Identification and Distribution among the Three Groups in the CVC and PICC Cohorts

Microbial identification and distribution in PICCs and CVCs are presented in Table 4. The five most common microorganisms isolated from total patients were MDR *Acinetobacter baumannii* (*n* = 47, 22.7%), MDR *Klebsiella pneumoniae* (*n* = 33; 15.9%), *Candida non-albicans* (*n* = 25, 12.1%), MDR *Pseudomonas aeruginosa* (*n* = 15; 7.2%), and non-MDR *Pseudomonas aeruginosa,* sharing equal numbers with *E. coli* (*n* = 10; 4.8%). 

The microorganisms isolated from colonized CVCs were Gram-negative bacteria (*n* = 112; 77.7%), Gram-positive bacteria (*n* = 14; 9.7%), and fungi (*n* = 17; 11.8%). The microorganisms isolated from colonized PICCs were Gram-negative bacteria (*n* = 24; 38.1%), Gram-positive bacteria (*n* = 15; 23.8%), and fungi (*n* = 24; 38.1%). Within the CVC group, the most common microorganism isolated was MDR *Acinetobacter baumannii* (*n* = 44; 30.6%), followed by MDR *Klebsiella pneumoniae* (*n* = 27; 18.7%). In the PICC group, the predominant microorganism isolated was *Candida non-albicans* (*n* = 15; 23.8%), followed by *Candida albicans*, coagulase-negative staphylococci, and MDR *Klebsiella pneumoniae* in equal numbers (*n* = 6; 9.5%). 

## 4. Discussion

Infections related to intravenous catheters significantly contribute to prolonged hospital stays, increased morbidity and mortality, and escalating healthcare costs. Therefore, it is crucial for researchers to investigate the risk factors to identify ways to reduce the incidence and impact of these serious complications on patients [19]. Many clinical studies frequently use catheter tip colonization as a surrogate endpoint for catheter-related bloodstream infections [20,21]. Specifically, short-term CVC-associated bacteremia most commonly originates from bacteria colonizing the catheter insertion site and migrating along the outer surface of the catheter into the bloodstream. In contrast, the intraluminal route (e.g., via the catheter hub) is considered more common for catheter colonization in long-term catheters [22,23]. 

To the best of our knowledge, our study is the first to examine CVCs and PICCs in terms of the time required for them to become colonized by common microorganisms and MDROs. According to our findings, the overall colonization rate of PICCs was noticeably lower than CVCs, despite the longer placement of the former. In CVCs, a high colonization rate initially occurred during the time period of 8–14 days after catheter insertion (group 2) and remained high in the following period. In contrast, for PICCs, the highest colonization rate appeared later CVCs, during the period from 2 to 3 weeks after catheterization (group 3). This pattern was also evident in MDROs rates, with the highest values observed in group 2 of the CVC and group 3 of the PICC cohorts. Moreover, the prevalence of MDROs was notably higher than non-MDROs in the CVC cohort. Conversely, in the PICC cohort, non-MDROs were the predominant category of isolates. 

Findings from previous studies addressing the association between catheter indwelling time and the risk of developing infections are contradictory. While some researchers argue in favor of an increased risk of infection over time [24,25], others do not believe that these infections are influenced by the indwelling time of catheterization [26,27]. We cannot directly compare our results with these studies, as the reported associations mainly pertain to CLABSIs rather than colonization events. Furthermore, most studies on catheter colonization focus on CVCs rather than PICCs [28,29]. Additionally, the patient populations examined in other studies consist of specific subgroups that are highly heterogeneous, making it challenging to draw comparisons with our study population [30].

Regarding the microbial distribution of our colonized catheters, our findings revealed that the majority of isolated microorganisms in CVCs were MDR *Acinetobacter baumannii*. This contrasts with most previous studies, which report Gram-positive bacteria, primarily *Staphylococcus* spp., as accounting for most colonization events, followed by Gram-negative bacteria and *Candida* spp. [31,32]. This discrepancy may be justified by existing differences in geographical regions, antimicrobial resistance epidemiology, and hospital environments worldwide [33]. Additionally, the isolation of MDR Gram-negatives aligns with the global increase in rates of MDROs in hospitals, especially ICUs.

On the contrary, in the PICC cohort, the bacterial distribution presented a different profile, with *Candida* spp. being the predominant isolate. This could be attributed to the longer indwelling time in patients with PICCs compared to those with CVCs [34]. *Candida* spp. has been reported as one of the most frequent opportunistic microorganisms causing nosocomial bloodstream infections over the last two decades [35]. 

The present study has specific limitations that need addressing. The retrospective data analysis of the two patient populations may have potential selection bias concerning different patient demographic characteristics, clinical signs and symptoms, and various therapeutic schemes. However, according to the data presented in Table 1, there were no significant differences in the mentioned characteristics. This is reasonable, as the indications for catheter use are the same for both types in our hospital—specifically, only for critically ill patients—to ensure patient safety and proper management during hospitalization. Additionally, there was no difference in APACHE score between the two groups, a variable that we consider the most straightforward indicator of illness severity.

## 5. Conclusions 

The results of the current study indicate that colonization and the rates of MDROs in PICCs were significantly lower compared to CVCs, despite the longer indwelling time of their placement. Additionally, the colonization by microorganisms, particularly MDROs, occurred later during the course of catheterization in PICCs compared to CVCs. This suggests that in terms of vascular infections, PICC lines may be a safer option than conventional CVCs for long-term intravenous access.

Our findings underscore the need for enhanced control and the implementation of measures to prevent intravenous catheter colonization in all hospital units, particularly in the ICU. Moreover, considering the notably high MDRO rates in microbial colonization patterns, we should thoroughly assess the clinical significance of catheter colonization through additional multicenter clinical studies. This will help strengthen the monitoring of catheterization and update the current guidelines based on evidence-based decision making.

## Figures and Tables

**Figure 1 antibiotics-13-00089-f001:**
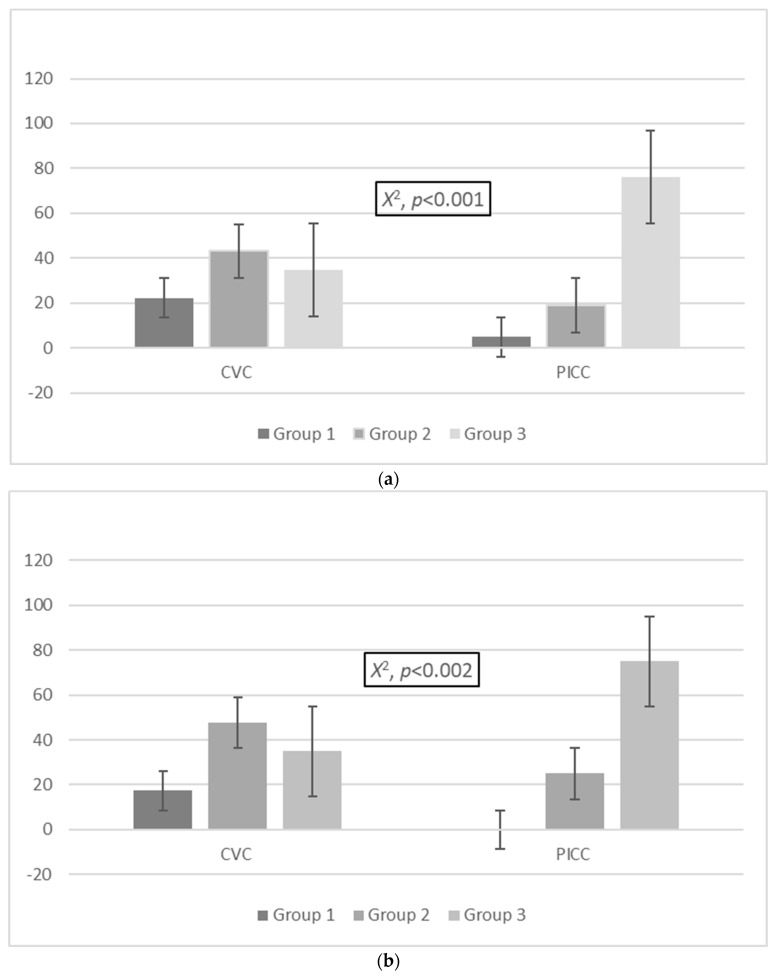
Distribution of (**a**) colonization events and (**b**) MDROs and (**c**) non-MDROs among the groups in the CVC and PICC cohorts. Group 1: ≤7 days, group 2: 8–14 days, group 3: >14 days. Abbreviations: MDROs, multidrug-resistant organisms; PICCs, peripherally inserted central venous catheters; CVCs, central venous catheters.

**Figure 2 antibiotics-13-00089-f002:**
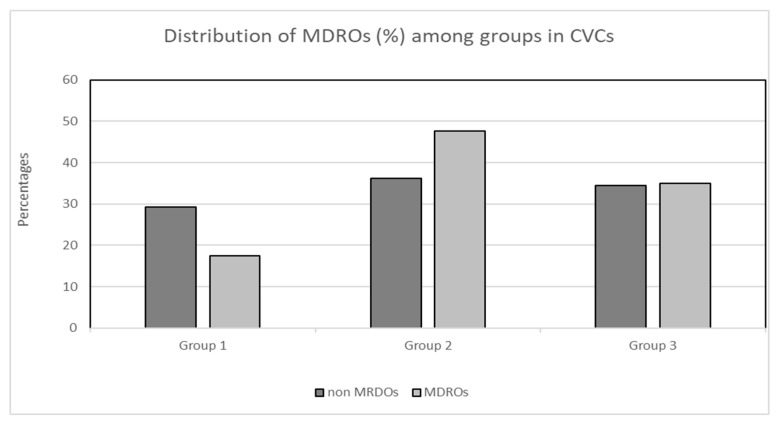
Distribution (%) of MDROs and non-MDROs among groups in the CVC cohort. Group 1: ≤7 days, group 2: 8–14 days, group 3: >14 days. Abbreviations: MDROs, multidrug-resistant organisms; CVCs, central venous catheters.

**Figure 3 antibiotics-13-00089-f003:**
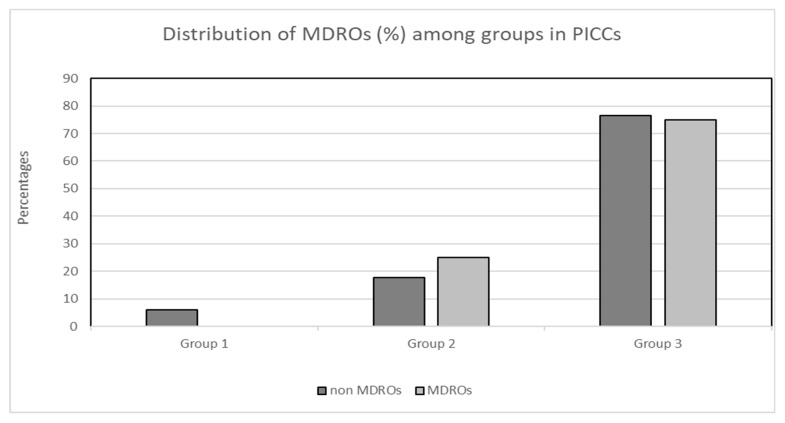
Distribution (%) of MDROs and non-MDROs among the three groups in the PICC cohort. Group 1: ≤7 days, group 2: 8–14 days, group 3: >14 days. Abbreviations: MDROs, multidrug-resistant organisms; PICCs, peripherally inserted central venous catheters.

**Table 1 antibiotics-13-00089-t001:** Study populations’ demographic and clinical characteristics.

Variable	N of Patients (%)		
	PICCs (*n* = 63)	CVCs (*n* = 144)	*p*-Value *
Age, mean ± SD, (years)	53.90 ± 17.98	55.62 ± 18.36	NS
Gender (M/F)	41/18	80/64	NS
Obesity	25 (39.6)	59 (41)	NS
Diabetes mellitus	15 (23.8)	31 (21.5)	NS
Pulmonary disease	8 (12.7)	18 (12.5)	NS
Hypertension	35 (55.5)	70 (48.6)	NS
Renal disease	10 (15.8)	27 (18.7)	NS
Oncologic disease	12 (19)	25 (17.3)	NS
Immune deficiency/suppression	18 (28.5)	39 (27)	NS
Admission category			
Medical	40 (63.5)	85 (59)	NS
Surgery	23 (36.5)	59 (41)	NS
Mechanical ventilation	7 (11.1)	15 (10.4)	NS
Cardiovascular disease	23 (36.5)	40 (27.7)	NS
Neurological disease	23 (36.5)	45 (31.2)	NS
Gastroenterological disease	15 (23.8)	38 (26.3)	NS
Hospital death	2 (3.1)	4 (2.7)	NS
APACHE score	13.22 ± 3.48	13.18 ± 3.1	NS

* A *p*-value of <0.05 was considered statistically significant. Abbreviations: SD, standard deviation; M/F, male/female; N, number; NS, not significant; PICCs, peripherally inserted central venous catheters; CVCs, central venous catheters.

**Table 2 antibiotics-13-00089-t002:** Colonization and MDRO incidence rates among groups in the CVC cohort.

Variables	≤7 Days	8–14 Days	>14 Days	Total	*p*-Value *
No. of catheters	904	202	81	1187	
Catheter days	4585	2874	2315	9774	
Colonization (No)	32	62	50	144	
Colonization (%)	3.5	30.7	61.7	12.1	
Colonization incidence rate (per 1.000 catheter/days)	6.98	21.57	21.60	14.73	0.019
MDROs (No.)	15	41	30	86	
MDROs (%)	1.7	20.3	37.0	7.2	
MDROs incidence rate (per 1.000 catheter/days)	3.27	14.27	12.96	8.79	0.025
non-MDROs (No.)	17	21	20	58	
non-MDROs (%)	1.9	10.4	24.7	4.8	
non-MDRO incidence rate (per 1.000 catheter/days)	3.71	7.31	8.64	5.93	0.5

* A *p*-value of <0.05 was considered statistically significant. Abbreviations: MDROs, multidrug-resistant organisms; No., number; CVCs, central venous catheters.

**Table 3 antibiotics-13-00089-t003:** Colonization and MDRO incidence rates among groups in the PICC cohort.

Variables	≤7 Days	8–14 Days	>14 Days	Total	*p*-Value *
No. of catheters	86	378	175	639	
Catheter days	2014	3756	5340	11,110	
Colonization (No.)	3	12	48	63	
Colonization (%)	3.49	3.17	27.43	9.8	
Colonization incidence rate (per 1.000 catheter/days)	1.49	3.19	8.99	5.67	0.047
MDRO (No)	0	3	9	12	
MDRO (%)	0.0	0.8	5.1	1.8	
MDRO incidence rate (per 1.000 catheter/days)	0.0	0.8	1.69	1.08	0.78
non-MDRO (No.)	3	9	39	51	
non-MDRO (%)	3.5	2.4	22.3	7.9	
non-MDRO incidence rate (per 1.000 catheter/days)	1.49	2.40	7.30	4.59	0.054

* A *p*-value of <0.05 was considered statistically significant. Abbreviations: MDROs, multidrug-resistant organisms; No., number; PICCs, peripherally inserted central venous catheters.

**Table 4 antibiotics-13-00089-t004:** Microbial identification and distribution in the PICC and CVC cohorts.

Microorganisms	CVCs (Νo. = 144)	PICCs (Νo. = 63)	Total
*Gram-negative bacilli*			
*Esherichia coli*	10 (6.9%)	0 (0.0%)	10 (4.8%)
*Enterobacter cloacae*	2 (1.4%)	3 (4.8%)	5 (2.4%)
*Klebsiella pneumoniae*	3 (2.1%)	3 (4.8%)	6 (2.9%)
*Morganella morganii*	1 (0.7%)	0 (0.0%)	1 (0.5%)
*Pseudomonas aeruginosa*	7 (4.9%)	3 (4.8%)	10 (4.8%)
*Proteus. mirabilis*	3 (2.1%)	0 (0.0%)	3 (1.4%)
*Serratia marcescens*	3 (2.1%)	3 (4.8%)	6 (2.9%)
*MDR Acinetobacter baumannii*	44 (30.6%)	3 (4.8%)	47 (22.7%)
*MDR Klebsiella pneumoniae*	27 (18.7%)	6 (9.5%)	33 (15.9%)
*MDR Pseudomonas aeruginosa*	12 (8.3%)	3 (4.8%)	15 (7.2%)
*Gram-positive cocci*			
CNS	3 (2.1%)	6 (9.5%)	9 (4.3%)
MRSA	3 (2.1%)	0 (0.0%)	3 (1.4%)
*Enterococcus faecalis*	2 (1.4%)	3 (4.8%)	5 (2.4%)
*Staphylococcus aureus*	1 (0.7%)	3 (4.8%)	4 (1.9%)
*Staphylococcus haemolyticus*	4 (2.8%)	3 (4.8%)	7 (3.4%)
*Streptococcus mitis*	1 (0.7%)	0 (0.0%)	1 (0.5%)
*Fungi*			
*Candida albicans*	7 (4.9%)	6 (9.5%)	13 (6.3%)
*Candida non-albicans*	10 (6.9%)	15 (23.8%)	25 (12.1%)
Other fungi	0 (0.0%)	3 (4.8%)	3 (1.4%)
*Other*			
*Bacillus* spp.	1 (0.7%)	0 (0.0%)	1 (0.5%)
Total	144	63	207

Abbreviations: PICCs, peripherally inserted central venous catheters; CVCs, central venous catheters; No., number; CNS, coagulase-negative staphylococci; MRSA, methicillin-resistant *Staphylococcus aureus;* MDR, multidrug-resistant; spp., subspecies.

## Data Availability

Data are contained within the article.

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
