# Peer review of "Central Venous Catheters versus Peripherally Inserted Central Catheters: A Comparison of Indwelling Time Resulting in Colonization by Multidrug-Resistant Pathogens"

_antibiotics, 2024, doi:10.3390/antibiotics13010089_

Round 1

Reviewer 1 Report

Comments and Suggestions for Authors

The topic of the manuscript is very interesting, and the presentation is very good. The authors can see some comments on the attached manuscript. 

Reviewer 2 Report

Comments and Suggestions for Authors

The authors have conducted a retrospective observational study in order to evaluate endovascular infections in central and peripheral catheters of hospitalized patients. The study is interesting and provides valuable information on nosocomial infections associated with catheters. The following modifications are suggested:

-The objective of the study is worded differently in Abstract and Introduction. It is suggested to unify the objective.

-Abstract: it is recommended to clarify that The study is limited to vascular infections (not infectious complications).

-Page 5: correct table 3

-Figures must be able to be read independently of the main text. MDRO, CVC, PICC, and groups 1, 2 and 3 remain to be explained.

-Central and peripheral catheters have different indications for clinical use, depending on the hospitalization ward, the reason for hospitalization, the type of medication to be administered or whether close monitoring of vital signs is needed. However, it would be necessary to clarify whether the PICCs were placed and changed during the stay or, on the contrary, the PICC remained in place throughout the entire hospitalization period.

I suggest clarifying that the study is limited to vascular infections. It is not appropriate to talk about infections in general or infectious complications (abstract). For example" in terms of vascular infections, PICC lines...."

Reviewer 3 Report

Comments and Suggestions for Authors

The authors provide important and interesting findings on microbial colonization of CVCs and PICCs. They note higher colonization rates in CVCs compared to PICCs despite the longer dwelling times of PICCs. Also, they note a tendency for CVCs to become colonized much early following insertion compared to PICCs. I note several inconsistencies in the results and the discussion sections of the manuscript and suggest the authors address them.

-         Firstly, the authors mention colonization rates and colonization events, these terms are used interchangeably and causes confusion to the reader. I would think colonization rates means incidence rates per 1000 catheter days. However, in the results section (specifically section 3.5) the heading says - “Comparison of distribution of colonization rates among the 3 groups…” - but the results discussed under this heading are colonization events (proportion of colonized catheters in each group in the CVCs and PICC cohorts). Again, in the next paragraph under 3.5 - when comparing the MDRO colonization events between groups the word MDRO rates is used. I would recommend a clear distinction be made about rates and events/proportion colonized.

-        In the same section 3.5, it appears the comparison was made between groups 1,2 and 3 in the CVC and PICC cohorts and not between CVCs and PICCs as noted in the heading and body of this section. The heading of 3.6 correctly points out the comparison is between groups and not between CVCs and PICCs.

-        In the body of section 3.6, it appears the results for MRDO and non-MRDOs are flipped. For instance, in group 1 of PICC cohort there were 0 MRDOs and 3 non-MDROs but it is mentioned that group 1 PICC has 3 MRDOs and 0 non-MDROs. The results in the figure and table are correct.

-        In the second paragraph of the discussion section the word colonization rate is used when talking about colonization events or the proportion colonized – “ Moreover, in CVCs the highest colonization rate of 43.1% presented in the time period …”.

-        - Also, in this paragraph the word significant is used multiple times. Does this mean statistically significant or just means that numbers were higher when comparing two entities. For instance, it is mentioned thatRemarkably, in CVCs’ cohort the rates of MDROs were significantly higher than the rates of non-MDROs …”. Here the word significant is used but I don’t see any analysis to directly compare colonization rates in MDROs and non-MDROs i.e. no p value.

-        -Also please check the accuracy of the last part of the sentence (in bold) “ It is worth noting that in the CVCs’ cohort, during the period of 8 to 14 days after catheter insertion not only was the colonization rate higher compared to the rest of the period, but it was also associated with a significant prevalence of MDROs over non-MDROs.” It says there is significant prevalence of MDROs over non-MRDOs but the analysis showed no significant difference for the CVC cohort as mentioned in section 3.6.

Reviewer 4 Report

Comments and Suggestions for Authors

Nice work, even though showing only comparably low numbers. What I personnally find very irritiating is that you published the exactely same work (just for another time period) in Antimicrobial Resistance and Infection Control a few months back. 

In Table 1, the gender data for CVC is missing.

Table 2: The title of the table is missing

Comments on the Quality of English Language

There are some grammatical errors, especially with regard to punctuation. 

Author Response

Reviewer #4

Nice work, even though showing only comparably low numbers. What I personnally find very irritiating is that you published the exactely same work (just for another time period) in Antimicrobial Resistance and Infection Control a few months back. 

>> Thank you for positive consideration of our study. Our previous manuscript (Pitiriga V. et al. Comparison of microbial colonization rates between central venous catheters and peripherally inserted central catheters. Antimicrob Resist Infect Control 2023, 12:74) specifically addresses colonization rates in CVCs and PICCs and the identification of pathogens, without investigating the duration required for these two types of catheters to be colonized. The present manuscript serves as a sequel to our previous publication, providing additional information. We acknowledge that this might not be evident from the abstract or the introduction section, and, we have included a paragraph in introduction section for reader clarity. Please see lines 85-94.

In Table 1, the gender data for CVC is missing.

>>The gender data in Table 1 was filled. Please see table 1.

Table 2: The title of the table is missing

>>The legend of Table 2 was present in the original manuscript, however not included in the pdf form. In the revised manuscript the legend is the following: Table 2. Colonization and MDROs incidence rates among groups in CVCs’ cohort. Please see table 2.

There are some grammatical errors, especially with regard to punctuation. 

>> We have corrected as many grammatical errors as possible throughout the entire manuscript.

Round 2

Reviewer 4 Report

Comments and Suggestions for Authors

Thank you for the revision of the manuscript. Especially the paragraph in the introduction part is very helpful.

Comments on the Quality of English Language

The English is fine